# Screening of Lignocellulolytic Enzyme Activities in Fungal Species and Sequential Solid-State and Submerged Cultivation for the Production of Enzyme Cocktails

**DOI:** 10.3390/polym13213736

**Published:** 2021-10-28

**Authors:** Nenad Marđetko, Antonija Trontel, Mario Novak, Mladen Pavlečić, Blanka Didak Ljubas, Marina Grubišić, Vlatka Petravić Tominac, Roland Ludwig, Božidar Šantek

**Affiliations:** 1Laboratory for Biochemical Engineering, Industrial Microbiology and Malting and Brewing Technology, Department of Biochemical Engineering, Faculty of Food Technology and Biotechnology, University of Zagreb, 10000 Zagreb, Croatia; nmardetko@pbf.hr (N.M.); atrontel@pbf.hr (A.T.); mnovak@pbf.hr (M.N.); mpavlecic@pbf.hr (M.P.); bljubas@pbf.hr (B.D.L.); mgrubisic@pbf.hr (M.G.); vpetrav@pbf.hr (V.P.T.); 2Biocatalysis and Biosensing Laboratory, Department of Food Science and Technology, BOKU—University of Natural Resources and Life Sciences, 1190 Vienna, Austria; roland.ludwig@boku.ac.at

**Keywords:** different fungal species, lignocellulolytic enzymes, *Fusarium* sp., pretreated corn cobs, sequential solid state and submerged cultivation, single and mixed fungal species cultivation

## Abstract

Various fungal species can degrade lignocellulolytic materials with their enzyme cocktails composed of cellulolytic and lignolytic enzymes. In this work, seven fungal species (*Mucor indicus* DSM 2185, *Paecilomyces variotii* CBS 372.70, *Myceliophthora thermophila* CBS 663.74, *Thielavia terrestris* CBS 456.75, *Botryosphaeria dothidea* JCM 2738, *Fusarium oxysporum* f.sp. *langenariae* JCM 9293, and *Fusarium verticillioides* JCM 23107) and four nutrient media were used in the screening for effective lignocellulose degrading enzymes. From the seven tested fungi, *F. oxysporum* and *F. verticilliodes*, along with nutrient medium 4, were selected as the best medium and producers of lignocellulolytic enzymes based on the determined xylanase (>4 U mg^−1^) and glucanase activity (≈2 U mg^−1^). Nutrient medium 4 supplemented with pretreated corn cobs was used in the production of lignocellulolytic enzymes by sequential solid-state and submerged cultivation of *F. oxysporum*, *F. verticilliodes*, and a mixed culture of both strains. *F. oxysporum* showed 6 times higher exoglucanase activity (3.33 U mg^−1^) after 5 days of cultivation in comparison with *F. verticillioides* (0.55 U mg^−1^). *F. oxysporum* also showed 2 times more endoglucanase activity (0.33 U mg^−1^). The mixed culture cultivation showed similar endo- and exoglucanase activities compared to *F. oxysporum* (0.35 U mg^−1^; 7.84 U mg^−1^). Maximum xylanase activity was achieved after 7 days of cultivation of *F. verticilliodes* (≈16 U mg^−1^), while *F. oxysporum* showed maximum activity after 9 days that was around 2 times lower compared to that of *F. verticilliodes.* The mixed culture achieved maximum xylanase activity after only 4 days, but the specific activity was similar to activities observed for *F. oxysporum.* It can be concluded that both fungal strains can be used as producers of enzyme cocktails for the degradation of lignocellulose containing raw materials, and that corn cobs can be used as an inducer for enzyme production.

## 1. Introduction

Globalization and rapid advances in technological development have positive effects on society (exchange of information, capital and labour mobility, open markets), but can have a negative impact on the environment and climate; therefore, there is a necessity for the constant optimization and improvement of the biotechnological production of new materials, biochemicals, and biofuels. One of the solutions for sustainable biotechnological development is the biorefinery concept. The biorefinery concept is based on the usage of various agricultural and industrial by-products as raw materials for the production of different bio-products. Moreover, it is necessary that such bioprocesses do not compete with the food and feed production [1]. The usage of residual lignocellulosic biomass as a feedstock for biofuels production, in the long term, can be an economically and environmentally viable solution if the cost-effective conversion of lignocellulosic biomass to fermentable sugars is achieved using the synergistic action of multiple enzymes [2].

The main sources of fermentable sugars in lignocellulosic biomass are cellulose and hemicellulose, which are very resistant to microbial degradation because of their complex and compact structures [3,4] with few access sites available for enzyme binding. The complex structure of lignocellulosic biomass necessitates an appropriate selection of pretreatment method [1,2,3,4]. For example, thermo-chemical pretreatment breaks the rigid structure of lignocellulose by disrupting the lignin coating and therefore enables subsequent enzymatic processing of polysaccharides [5,6]. To efficiently hydrolyze cellulose, at least three enzymes are required to work in synergy: endoglucanases, which randomly cleave polymer bonds; exoglucanases, which remove mono- and dimers from the ends of the glucose chain; and *β*-glucosidase, which hydrolyzes cellobiose, yielding two molecules of glucose [4,6,7]. For the hydrolysis of hemicellulose, enzymes with activity towards diverse hemicellulosic polymers are required, and these enzymes must deal with the greater structural complexity of hemicelluloses compared to cellulose. The best-studied enzyme is xylanase, which catalyzes the hydrolysis of the glycosidic *β*-1-4 linkage of xylosides. While the polysaccharides in lignocellulose are mainly degraded by various hydrolases, lignin degradation is an oxidative process in which phenol oxidases are key enzymes. Of these, lignin peroxidases (LiP), manganese peroxidases (MnP), and laccases have been best studied. There are also other enzymes that participate in the lignin degradation processes. They are various flavin-containing oxidases that are H_2_O_2_-dependent, and which can be located either intra- or extracellularly [3,8].

One of the good examples of lignocellulolytic enzymes (e.g., cellulases, xylanases, or laccases) application is in the food production. In the food industries, lignocellulolytic enzymes are often used during the processing of fruits and vegetables, texturizing and flavoring of food products, juice processing, oil extraction, and alcoholic beverage production. Extensive use of these enzymes in different food processing industries not only accelerates the food production, but also improves product quality [4,7].

The most important and widespread group of organisms which contribute to the enzymatic degradation of lignocellulolytic materials are fungi or bacteria, which can be isolated from different natural habitats, such as soil, crop residues, compost, tree bark, and insects, or by using a metagenomic approach [9]. Some known fungi are *Trichoderma reesei*, *Pleurotus ostreatus*, *Tremetes versicolor*, *Neurospora crassa*, *Aspergillus oryzae*, etc. [10,11,12]. These filamentous fungi secrete different enzymes depending on the morphological features of the microorganism and environmental factors. Different growth morphologies are the result of various culture conditions which in the end affect enzyme production [13]. Several studies have investigated the influence of some conditions on enzyme yields. Domingues et al. [14] and An et al. [15] researched the influence of the type of nutrient medium, Sohail et al. [16] focused on the influence of temperature and pH, while Ahamed and Vermette [17] researched mixed culture cultivations.

When fungi are grown on different lignocellulosic substrates, their crude enzyme extracts contain various protein profiles, which is directly influenced by the choice of the inducer substrate. Studies have shown that by using lignocellulosic substrates, instead of commercial inducers like lactose or carboxymethyl cellulose, cellulase production is affected and results in specific protein profiles and improved hydrolysis yields [18,19]. Lignocellulolytic enzymes are currently produced by using either solid-state fermentation or submerged fermentation [20,21]. However, there have been few reports on lignocellulolytic enzyme production using sequential solid state and submerged fermentation [22]. Enzyme cocktails produced using these methods are more efficient in hydrolyzing plant structures, due to the synergistic effect, where one enzyme can act on the product of another [23]. Processes employing enzyme blends are also more efficient and ecologically friendly compared to chemical processes used in the degradation of lignocellulose, since it is possible to adjust the enzyme types and loads to fully convert each substrate. Enzyme blends should be designed to accommodate the specific substrate, since every substrate possesses unique characteristics and demands the right conditions for its total conversion to simple compounds. The construction of efficient enzymatic cocktails requires the investigation of fungi growth patterns on various lignocellulosic materials, and the identification of the types of enzymes they produce [24].

The aim of this research was to screen available fungi strains for lignocellulolytic enzyme activities that can break down pretreated lignocellulosic biomass into simple sugars, and to investigate the impact of four different types of nutrient medium on enzyme yields. Subsequently, the most promising strains and medium were used to produce enzyme cocktails and to evaluate a novel sequential cultivation method for lignocellulolytic enzyme cocktails production. This technique is based on dilute acid pretreated corn cobs as the inducer substrate for pre-culture cultivation, using solid state followed by submerged cultivation. Afterwards, LC-ESI-MS was used to identify the enzymes present in the cocktails.

## 2. Materials and Methods

### 2.1. Experimental Setup

In this study, seven fungal strains were screened for cellulolytic enzyme activities by using a plate assay in carboxymethyl cellulose (CMC; Sigma Aldrich CHEMIE, Steinheim, Germany) agar using Congo red (Sigma Aldrich, St. Louis, MO, USA) as the dye. Afterwards, screening experiments for lignocellulolytic and starch degrading enzyme activity were undertaken by using sequential solid state and submerged cultivation containing pretreated corn cobs as the inducer substrate. Scale up of cultivations by selected strains was performed, and enzyme activities were monitored for 14 days. The obtained supernatants were used for partial purification and LC-ESI-MS (maXis 4G ETD; Bruker, Billerica, MA, USA) was used to identify the enzymes present in the cocktails.

### 2.2. Feedstock

Dilute acid pretreated corn cobs (92.61% dry weight) obtained from the Laboratory for Biochemical Engineering, Industrial Microbiology and Malting and Brewing Technology (Faculty of Food Technology and Biotechnology, University of Zagreb) were used as a feedstock. Corn cobs were reduced in size by using hammer mill (NA45 Megametal d.o.o., Nedelišče, Croatia) and sieved through a frame size of 5 mm. Particles with a diameter less than 5 mm were used for pretreatment in high pressure reactor as described in Marđetko et al. [25]. The pretreated corn cobs composition was determined by the NREL method [26]: 55.58% glucans, 3.11% xylans, 0.48% arabinans, 4.35% acetic acid, 4.31% formic acid, 0.21% acid soluble lignin, 27.3% acid insoluble lignin and ash, and 4.6% (*w*/*w*) other constituents.

### 2.3. Working Microorganisms and Media

In this study, fungi provided by the RikenBRC (Kyoto, Japan) through the National BioResource Project of the MEXT/AMED Japan (*Botryosphaeria dothidea* JCM 2738 (BD), *Fusarium oxysporum* f.sp. *langenariae* JCM 9293 (FO), and *Fusarium verticillioides* JCM 23107 (FV)) from the culture collection of the Laboratory for Biochemical Engineering, Industrial Microbiology and Malting and Brewing Technology (Zagreb, Croatia) (*Mucor indicus* DSM 2185 (MI)) and the CBS Fungal Diversity Centre and Technology (Utrecht, The Netherlands) (*Paecilomyces variotii* CBS 372.70 (PV), *Myceliophthora thermophila* CBS 663.74 (MT), and *Thielavia terrestris* CBS 456.75 (TT)), were used as working microorganisms. Cultures were maintained on standard potato dextrose agar (Difco, BD, Franklin Lakes, NJ, USA).

The media composition used for the cultivations of fungi are given in Table 1. All chemicals used in media preperation were acquired from Sigma Aldrich, St. Louis, MO, USA, if not stated otherwise. The pH value of the media was set to a value of 6 with the addition of NaOH granules (Sigma Aldrich, St. Louis, MO, USA), and afterwards, the media were sterilized at 121 °C for 15 min. The salt solution was filtered through a disposable sterile syringe filter (CHROMAFIL Xtra PA-20/25, Duren, Germany) and added to the media before use (Table 1). 

### 2.4. Cellulolytic Enzyme Activity Plate Test

The fungi for the cellulolytic enzyme activity test were precultured on potato dextrose agar (Difco, BD, Franklin Lakes, NJ, USA) at 30 °C for 5 days. For the observation of fungal cellulolytic enzyme activity, the precultures were transferred onto media containing (in g L^−1^) 5 carboxymethyl cellulose (Sigma Aldrich, St. Louis, MO, USA); 1 yeast extract (Difco, BD, Franklin Lakes, NJ, USA); 0.5 Congo red (Sigma Aldrich, St. Louis, MO, USA); and 15 agar powder (Sigma Aldrich, St. Louis, MO, USA). After 7 days of culturing at 30 °C, evaluation of enzyme activity was performed by observing a clear zone formed around the fungal colony. The enzymatic index (EI) was calculated by measuring the clearance zone and using Equation (1) [27,28]:EI = diameter of hydrolysis zone/diameter of colony(1)

### 2.5. Preculture Cultivation for the Induction of Lignocellulolitic Enzyme Production

The preculture of all seven fungi was initiated as solid-state fermentation, using dry dilute acid pretreated corn cobs as the solid substrate. The substrate was moisturized by adding 20 mL of sterilized nutrient medium into a 500 mL baffled flask containing 5 g of dry pretreated corn cobs. A volume of spore suspension resulting in a concentration of 10^6^ spores per gram of dry substrate was added. Cultivations were carried out at 30 °C for 24 h under static conditions. After that time period, 200 mL of each corresponding nutrient medium shown in Table 1, enriched with 30 g L^−1^ of glucose, was added to the analogous flasks and the cultivation was continued as submerged cultivation at 30 °C for 48 h under continuous agitation at 180 rpm. A volume of each preculture suspension (MI, PV, MT, TT, BD, FO, and FV) corresponding to 10% (*v*/*v*) was transferred to initiate lignocellulolytic enzyme production in shake flasks. In the case of the mixed culture cultivation (FO and FV) where 5% (*v*/*v*) of each strain preculture was added.

### 2.6. Lignocellulolytic Enzyme Production

The culture media used for enzyme production were akin to those used for the precultures, with the exception of the addition of 10 g L^−1^ glucose and 2% (*w*/*v*) of dry pretreated corn cobs.

The evaluation of media composition on enzyme production was first estimated in shake flasks. The preliminary cultivations were performed for 96 h in 500 mL Erlenmeyer flasks, with a 100 mL working volume, at 30 °C and 180 rpm. The pH was set at 6.0 and monitored and corrected on a daily basis. Samples were collected after 96 h, centrifuged at 6000× *g* for 10 min, and the crude enzymatic extracts were used for quantification of cellulolytic enzyme activities and reducing sugars. The results obtained were analyzed and the fungi and medium with the highest enzyme activities were selected for further experiments.

Cultivations conducted in 1 L shake flasks, with 500 mL of media, were carried out for 14 days at 30 °C and 200 rpm. Samples were collected at 24 h intervals and centrifuged at 6000× *g* for 10 min. The crude enzymatic extract was used for the quantification of endoglucanase, exoglucanase, and xylanase activities, as well as the reducing sugars concentration.

The results showed that enzyme activity was the highest after 6 days, so the cultivations were repeated and stopped after 6 days. The fermentation broth was clarified by centrifugation (6000× *g*; 10 min) and the obtained crude extract was used for further protein purification and characterization. All cultivations were carried out in duplicate.

### 2.7. Protein Purification

Diafiltration was used to remove salts and small molecules from the broth. The clarified fermentation broth was purified, concentrated, and buffer exchanged to acetate buffer (pH value 6) by using a hollow fiber crossflow module (Microza UF module SLP-1053, 10 kDa cut-off; Pall Corporation, Port Washington, NJ, USA). The samples were stored at 4 °C, and later used for the determination of enzyme activity and SDS-PAGE.

### 2.8. Analytical Methods

#### 2.8.1. Cellulolytic and Hemicellulolytic Activity Assay

The reducing sugars concentration was determined by the 3,5-dinitrosalicylic acid (DNS) method [29]. The activity of several cellulolytic enzymes can be estimated by incubation of the sample solution with different cellulosic or hemicellulosic substrates and quantification of the liberated sugars (Table 2) by the standard method proposed by Ghose [30]. The reactions were recorded using a Lambda 35 UV/VIS spectrometer (Perkin Elmer, Waltham, MA, USA). One unit of enzymatic activity is determined as the amount of enzyme that releases 1 µmol of the corresponding reducing sugar per minute under the specified assay conditions. Specific enzyme activity was expressed as enzyme units per mg of total protein.

#### 2.8.2. Laccase Activity Assay

This method is based on the detection of the oxygen-dependant oxidation of 2,2′-azinobis-(3-ethylbenzthiazoline-6-sulfonic acid) (ABTS; Sigma Aldrich, St. Louis, MO, USA) to the ABTS cation radical [31]. The reaction solution was prepared by mixing 100 µL of 10 mM ABTS solution and 880 µL of 100 mM sodium-acetate buffer (pH 4.0) in a 1 mL microcuvette. The microcuvettes were incubated for 20 min in a 30 °C water bath, after which they were placed in sample holders. The reaction was started by adding 20 µL of the sample to the cuvette and recorded using a spectrometer (Lambda 35 UV/VIS, Perkin Elmer, Waltham, MA, USA) at 420 nm. One unit of enzyme activity was 1 µmol of ABTS cation radical produced per minute per mL of the supernatant. Specific enzyme activity was expressed as enzyme units per mg of total protein.

#### 2.8.3. Lignin Peroxidase (LiP) Assay

This method is based on the oxidation of veratryl alcohol(3,4-dimethoxybenzyl alcohol; Sigma Aldrich, St. Louis, MO, USA) to veratraldehyde [32]. The standard reaction mixture consisted of 1 mL of 125 mM sodium tartarate buffer (pH 3.0), 500 mL of 10 mM veratryl alcohol, 500 mL of 2 mM hydrogen peroxide solution, and 500 mL of the culture filtrate. The reaction was initiated by adding hydrogen peroxide (Sigma Aldrich, St. Louis, MO, USA) and the change in absorbance was monitored at 310 nm. One unit of enzyme activity was 1 µmol of veratraldehyde produced per minute per mL of the supernatant. Specific enzyme activity was expressed as enzyme units per mg protein.

#### 2.8.4. Cellobiose Dehydrogenase Assay

This method is based on the reduction of cytochrome *c* which changes the color from the initial orange to a more reddish pink tone [33]. A total of 20 µL of the cytochrome *c* (Sigma Aldrich, St. Louis, MO, USA) solution, 100 µL of 300 mM lactose, and 860 µL of 100 mM sodium-acetate buffer (pH 4.0) were pipetted into a clean cuvette. The cuvette was incubated for at least 20 min at 30 °C in a water bath, and afterwards placed into the sample holder. The reaction was started by adding 20 µL of the sample to the solution. The absorbance was recorded at 520 nm (Lambda 35 UV/VIS, Perkin Elmer, Waltham, MA, USA) for 120 s. Specific enzyme activity was expressed as enzyme units per mg of protein.

#### 2.8.5. Bradford Protein Assay

The samples were prepared by mixing 15 µL of the sample with 600 µL of the Bradford reagent (Bio-Rad, Hercules, CA, USA) in a plastic cuvette [34]. The cuvette was left in the dark, at room temperature, for 15 min, after which the absorbance of the samples was measured at 595 nm using a UV/VIS spectrophotometer (Beckman Coulter DU 800; Beckman Coulter, Brea, CA, USA). Protein concentrations were automatically calculated by the software.

#### 2.8.6. Sodium Dodecyl Sulphate Polyacrylamide Gel Electrophoresis (SDS-PAGE)

Before the electrophoresis was started, 20 µL of Laemmli sample buffer (Bio-Rad, Hercules, CA, USA) was added to the crude and partially purified samples. The samples were then heated by using a digital dry bath to 98 °C and were kept at that temperature for 5 min. For the electrophoresis, Mini-PROTEAN TGX Precast Gels (Bio-Rad, Hercules, CA, USA), Mini PROTEAN Tetra Cell (Bio-Rad, Hercules, CA, USA), and the PowerPac HC (Bio-Rad, Hercules, CA, USA) power supply were used. The buffer tank was filled with the reused 1× running buffer, while the reservoir between the gels was filled with a freshly prepared and 10 times diluted 10× running buffer (30.3 g L^−1^ TRIZMA base, 144 g L^−1^ glycine, 10 g L^−1^ SDS (Sigma Aldrich, St. Louis, MO, USA)). One well was used for 15 µL of the unstained Precision Plus Protein Standard (Bio-Rad, Hercules, CA, USA) while the other four wells were loaded with 15 µL of the denatured samples. The samples were run at 120 V for about an hour. After the electrophoresis was finished, the gels were stained using QC Colloidal Coomassie (Bio-Rad, Hercules, CA, USA) and washed using distilled water. A picture of the gel was taken using the Universal Hood II Gel Doc System (Bio-Rad, Hercules, CA, USA).

#### 2.8.7. Peptide Profiling by LC-ESI-MS

The LC-ESI-MS analysis was based on the previously described method [35] and was adapted to the given instrumentation. To this end, bands seen in the SDS-PAGE gel were cut out and digested with sequencing-grade chymotrypsin (Roche, Basel, Switzerland). The peptide mixture was analyzed using a Dionex Ultimate 3000 system linked to a quadrupole time of flight (Q-TOF) MS instrument (maXis 4G ETD; Bruker, Billerica, MA, USA) equipped with the standard ESI source in the positive ion, data-dependent acquisition (DDA) mode (switching to MS/MS mode for eluting peaks). MS scans were recorded (range, 150 to 2200 *m*/*z*; spectrum rate, 1.0 Hz) and the six highest peaks were selected for fragmentation (collision-induced dissociation (CID) mode). Instrument calibration was performed using an ESI calibration mixture (Agilent, Santa Clara, CA, USA). For separation of the peptides, a Thermo BioBasic C 18 separation column (5 m particle size, 150 by 0.320 mm) was used. A gradient from 97% solvent A and 3% solvent B (solvent A, 65 mM ammonium formate buffer, pH 3.0 (formic acid supplied by Carl Roth, Karlsruhe, Germany; ammonia supplied by VWR BDH Prolabo, Poole, UK); solvent B, 80 % acetonitrile (ACN; VWR BDH Prolabo, Poole, UK) and 20% solvent A) to 40% solvent B in 45 min was applied, followed by a 15-min gradient from 40% solvent B to 95% solvent B at a flow rate of 6 L min^−1^ at 32 °C. DataAnalysis 4.0 (Bruker, Billerica, MA, USA) was used for peptide evaluation.

#### 2.8.8. Statistical Analysis

The standard deviation of experimental data was calculated by using the standard procedure in the software Statistica 12.0 (StatSoft; Tulsa, OK; USA). Standard deviations are presented as bars in tables and figures.

## 3. Results and Discussion

### 3.1. Screening of Fungi for Lignocellulolytic Enzyme Activity

The conversion of cellulose and hemicellulose to simple fermentable sugars is caused by the synergistic actions of various enzymes that belong to a complex system of cellulase and hemicellulase enzymes. The enzymes most active in cellulose hydrolysis are endoglucanase, exoglucanase, and *β*-glucosidase, while the enzymes responsible for hemicellulose degradation are exoxylanase, endoxylanase, *β*-d-xylosidase, and others [3].

The preliminary evaluation of enzymatic activity was carried out using a plate assay in CMC agar using Congo red as the dye. The seven cultures that were submitted to this assay were *M. indicus*, *P. variotii*, *M. thermophila*, *T. terrestris*, *B. dothidea*, *F. oxysporum* f.sp. *langenariae*, and *F. verticillioides*. Based on the growth patterns, all fungi strains showed some degree of zone clearing, but the measured diameter of the mycelial growth area varied depending on the strain. The widest clear zone was seen in *F. oxysposrium* and *F. verticillioides*, with an EI value of 1.1 and 1.09, respectively, while the lowest EI value was observed with *M. indicus*, 1.01. All other strains had an EI value in between these highest and lowest values: *B. dothidea*, 1.07; *P. variotii*, 1.05; *M. thermophila*, 1.03; *T. terrestris*, 1.03. Similar values were reported for some of the fungi from the genera *Mucor*, *Penicillium*, *Paecilomyces*, *Fusarium*, *Aspergillus*, and *Trichoderma* or related strains in the literature [27,28,36,37]. These data suggest that the fungi can use CMC as the sole carbon source, which indicates the possibility that they may also be able to use cellulose.

Many studies on the effect of nutrient medium on enzyme production have been reported [9,15,38,39]. However, most studies have focused on the substitution of a nitrogen or carbon source, and fewer articles consider the effect of a lignocellulosic substrate as the enzyme producing inducer. Therefore, in this study four different media (Table 1) were tested by a simple comparison of specific enzyme activity. Positive cultures were grown in these different media with diverse compositions, supplemented with dry pretreated corn cobs, to investigate which medium is the best for the induction of enzyme production. A novel cultivation method named sequential solid state and submerged cultivation, which is reported to have higher yields of enzymes compared to solid state and submerged cultivations, was investigated [22].

Enzymatic production was carried out after the initial preculture cultivation. After 96 h of enzyme production at 30 °C, the cultivations were stopped, the media were centrifuged, and the crude enzyme extracts were used for enzyme identification (Table 2). Screening was targeted to the expected most important enzyme classes (endo- and exoglucanases; xylanases; mannanases; arabinases; pectinases) and on other enzymes that could be expressed and show activity on the chosen pretreated feedstock (cellobiose dehydrogenase (CDH); laccase and lignin peroxidase (LiP)) (Table 1). Cultivations were performed in duplicate, and differences smaller than 10% were observed for enzymatic activities.

#### 3.1.1. Hemicellulolytic Activity

Screening results for starch degrading, cellulolytic, and hemicellulolytic activities showed that all the tested fungi have some level of enzyme activities (Figure 1). The most important hemicellulose degrading enzymes are xylanases, arabinases, and mannanases [3]. Several studies have focused on the production of xylanases [40,41,42,43,44,45]. The most efficient xylanase producing strains in all the tested media were *F. oxysporum* f.sp. *langernariae* JCM 9293, *F. verticillioides* JCM 23107, and *B. dothidea* JCM 2738. Out of these fungi, the maximum xylanase activity was achieved by *F. verticillioides.* Average xylanase activities ranged from 2.53 U mg^−1^_total protein_ in Medium 4, to 6.48 U mg^−1^ in Medium 1, with small differences compared to Media 2 and 3 (6.35 U mg^−1^ and 6.18 U mg^−1^). When comparing these *F. verticillioides* activities with the ones reported by Saha [46], an increase can be noted. He reports an activity of 1.3 U mg^−1^ in the culture supernatant, when the culture was grown using corn fiber xylan as the carbon source. *F. oxysporum* and *B. dothidea* exhibited slightly lower activities, ranging from 2.25 U mg^−1^ to 2.88 U mg^−1^ and from 2.17 U mg^−1^ to 5.64 U mg^−1^, respectively. These activities are also higher than those reported by Arabi et al. [47], who evaluated the production of xylanase by 21 isolates of *Fusarium sp*. under solid state fermentation on agriculture wastes. In their research, the highest reported xylanase activity was achieved by *F. solani* SYRN7, 0.91 U mg^−1^, while *F. verticillioides* showed a much lower activity of 0.139 U mg^−1^_total protein_. *F. verticillioides*, *F. oxysporum*, and *B. dothidea* also showed the highest arabinase activities among the tested strains, with the maximum values reached in Medium 4, the lowest of 1.51 U mg^−1^ by *B. dothidea*, and the highest of 2.51 U mg^−1^ by *F. verticillioides*. Comparing mannanase activities, the best producer was *F. verticillioides*, 2.35 U mg^−1^, followed by *F. oxysporum*, 2.08 U mg^−1^.

#### 3.1.2. Cellulolytic Activity

The most important cellulose degrading enzymes are endo- and exoglucanase [3]. In previous years, several researchers investigated their production [48,49,50,51,52]. *F. oxysporum*, *F. verticillioides*, and *B. dothidea* also had the highest overall average exo- and endoglucanase activities, which are important in cellulose degradation. The highest endoglucanase activities were detected in Medium 4, varying from 0.51 U mg^−1^ for *B. dothidea* to 1.45 U mg^−1^ for *F. verticillioides*, while the lowest activities were detected in Medium 1. Exoglucanase activities were comparatively high in all the tested media, with a peak of 2.76 U mg^−1^ in Medium 2 in which *F. oxysporum* was cultivated, followed by Medium 4 in which activities of 1.83 U mg^−1^, 2.11 U mg^−1^, and 2.0 U mg^−1^ were measured for *B. dothidea*, *F. verticillioides*, and *F. oxysporum*, respectively. Ravalason et al. [53], investigated the secretome of *F. verticillioides* to enhance the saccharification of wheat straw. Their report coincides with our research regarding hemicellulases and pectinases activity shown by *F. verticillioides*, but no cellulose degrading enzymes were reported. In a similar research work, deb Dutta et al. [54] analyzed the cellulolytic activity of some *Fusarium* species based on the CMC assay, filter paper assay, or cotton assay. Different *F. oxysporum* strains showed high CMCase activity (0.445 IU ml^−1^) and FPase activities (9.25 IFPU ml^−1^).

*M. thermophila* CBS 663.74, *P. variotii* CBS 372.70, and *T. terrestris* CBS 456.75 showed some degree of enzyme activities, which were 2 to 10 times lower compared to those of *F*. *verticillioides*, *F. oxysporum*, and *B. dothidea*. This could be the result of inadequate cultivation conditions, in particular temperature, which was 30 °C, while these fungi have higher optimal growth temperatures [55,56,57]; the fungus with the overall lowest cellulolytic and hemicellulolytic activities compared to the other strains was *Mucor indicus* DSM 2185.

#### 3.1.3. Lignin Degrading Activity

Lignin degradation by fungi was carried out by oxidoreductases like phenol oxidases (laccase) or heme peroxidases (lignin peroxidase (LiP), manganese peroxidase (MnP) or versatile peroxidase (VP)). Out of all the tested fungi, some laccase activity was shown by *P. variotti*, *F. oxysporum*, *M. thermophila*, and *B. dothidea*, while LiP activity was detected in the supernatants of *F. verticillioides* and *P. variotii.* Even though these activities were detected, they were lower than 0.1 U mg^−1^, which is in accordance with data reported for other soft rot fungi like *Alternaria alternata*, *Penicillium chrysogenum*, *Botrytis cinerea*, and *Fusarium* sp. [58,59].

Overall, the fungi with the highest activities were *F. oxysporum* f.sp. *langernariae* JCM 9293, which exhibited the highest cellulolytic enzyme activities, and *Fusarium verticillioides* JCM 23107, which had the highest hemicellulolytic enzyme activities, cultivated in Medium 4, which contains corn steep liquor as the nitrogen source and was supplemented with pretreated corn cobs as the inducer substrate, as can be seen from Figure 1. Correspondingly, this medium and these strains were used in all subsequent experiments.

### 3.2. Sequential Solid State and Submerged Cultivation of Selected Fungi Strains

The most important enzymes in cellulose and hemicellulose degradation are endo- and exoglucanase and xylanase, as mentioned earlier. Pretreated corn cobs are a complex lignocellulosic material that have a high glucan and small xylan content and as such were used as the inducer of enzyme production in the sequential solid-state and submerged cultivation of the selected fungal strains [15,22,60,61,62,63,64,65]. The activity of glucanases and xylanases were monitored during the cultivation of *F. oxysporum* f.sp. *langernariae* JCM 9293 and *F. verticillioides* JCM 23107 in medium that was inoculated with only one species of selected fungal strain, and in a separate experiment with a mix of both species (each 5% *v*/*v*) of selected fungal strains. The cultivation of a mixed culture in the same media was conducted to research the possibility of increased glucanase and xylanase activity [51,52,53,54]. The cultivations were performed in duplicate, and differences smaller than 10% were observed for enzyme activities. The experimental results are shown in Figure 2.

During the cultivation of selected fungi, *F. oxysporum* and *F. verticillioides* grew in a filamentous morphological form that moderately increased broth viscosity, but it was still possible to maintain good recirculation and mixing of media [66]. As stated, xylanase and endoglucanase production during sequential cultivation seemed to be induced by the presence of the lignocellulosic biomass and the maximum determined enzymatic activities occurred when the added carbon source was exhausted. The added glucose in the fermentation broth for the cultivation of the selected fungi was decomposed after one day of cultivation (data not shown). Maximum exoglucanase activity during cultivation of *F. verticillioides* was achieved after 5 days of cultivation (Figure 2) and was 0.56 U mg^−1^_total protein._ In comparison, in the sequential solid state and submerged cultivation of *F. oxysporum*, maximum exoglucanase activity was determined to be more than 6 times higher (3.33 U mg^−1^). This activity was achieved after 7 days of cultivation. *F. oxysporum* also showed 2 times higher endoglucanase activity (0.33 U mg^−1^) in comparison with *F. verticillioides* (0.15 U mg^−1^). Maximum endoglucanase activities were achieved after 4 days of cultivation (Figure 2).

Xylanases are important in various industrial processes and are intensively studied. New bioprocesses for the effective production of xylanases are of great importance [65]. In all cultivations, the fungi produced xylanases to degrade hemicellulose still present in the pretreated corn cobs (Figure 2). Fungi *F. verticillioides* JCM 23107 showed almost 16 U mg^−1^ of xylanase activity after 7 days of cultivation, after which it started to decrease. Maximal xylanase activity during the cultivation of *F. oxysporum* f.sp. *langernariae* JCM 9293 was achieved after 9 days of cultivation and was 8.36 U mg^−1^, almost two times lower than in *F. verticillioides* cultivation. Higher xylanase activity the during sequential solid state and submerged cultivation of selected fungi strains is well known in the literature [15,22,65,67,68]. For this kind of research, qualitive and quantitative data comparison is challenging as different researchers show their enzyme activities in different manner [40,41,42,43,44,45,46,47,48,49,50,51,52,53,60,65,67,68].

As the selected investigated fungi showed different glucanase and xylanase activity during their sequential solid state and submerged cultivation in media containing pretreated corn cobs, it was interesting to see if total endoglucanase and xylanase activity could be increased with the combination of two selected fungi. The inoculum in this research was preculture suspension corresponding to 10% (*v*/*v*; 5% of each fungus) of *F. oxysporum* and *F. verticillioides*. In the cultivation with the mix of two selected fungi, maximal exoglucanase activity was achieved after 4 days of cultivation and was 3.35 U mg^−1^. In the same amount of time, fungi *F. oxysporum* showed 2.66 U mg^−1^ exoglucanase activity, that is, approximately 20% less, and *F. verticillioides* showed only 0.42 U mg^−1^ (almost 90% less). The endoglucanase activity in mixed culture cultivation was nearly the same as during cultivation of *F. oxysporum* (≈0.35 U mg^−1^) and almost two times higher than that of the cultivation of *F. verticillioides* in the same media. Xylanase activity was at its maximum after 4 days of the mixed culture cultivation and was 7.84 U mg^−1^ (Figure 2). In the same time, the fungi *F. verticillioides* showed an activity of 6.69 U mg^−1^, and the fungi *F. oxysporum* showed an activity of 2.86 U mg^−1^. These results with mixed culture cultivation show that this kind of cultivation does not significantly increase endoglucanase and xylanase activity. A probable reason could be the antagonistic microbial interaction of the two selected fungi as reported in several literature overviews [69,70,71].

#### Partial Purification and LC-ESI-MS Analysis

As shown in Figure 2, maximum glucanase and xylanase activity was detected after ~6 days of cultivations; the cultivations were repeated to obtain the fermentation broth used for further activity assays. Clarified fermentation broth-crude extract was concentrated using a hollow fiber crossflow module with a 10 kDa cut-off membrane. By this method, crude extract was partially purified and concentrated approximately 7.5- to 8-fold. Enzyme activities were determined for the crude and partially purified extract and are shown in Table 3. An increase of enzymatic activity after ultrafiltration was detected for all enzymes. Two-fold increase was detected for starch degrading, endo- and exoarabinase, exoglucanase, and xylanase activity for both *Fusarium* sp., while a 2.5- to 3-fold increase was detected for mannanase and exoglucanase. The highest increase in activity can be observed for pectinase and was 3.5 to 4-fold for both *Fusarium* sp. Relatively high xylanase and pectinase activities are important for the enzymatic hydrolysis of feedstocks with a high content of xylans and pectins rich in arabinose [3].

The crude extract and partially purified extract were used for SDS-PAGE to obtain a protein profile regarding molecular weight (Figure 3). Afterwards, selected bands in the gel were cut out and prepared for LC-ESI-MS analysis. Some of the identified enzymes present in the gel in varying concentrations depending on the producing organism were cellulase (E.C. 3.2.1.4), endo-1,3(4)-*β*-glucanase (E.C. 3.2.1.6), *β*-glycosidase (E.C. 3.2.1.21), licheninase (E.C. 3.2.1.73), α-galactosidase (E.C. 3.2.1.22), glucan 1,4-*α*-glucosidase (E.C. 3.2.1.3), endo-1,4-*β*-xylanase (E.C. 3.2.1.8), *β*-1,3-glucanosyltransferase (E.C. 2.4.1), α-*N*-arabinofuranosidase (E.C. 3.2.1.55), endopolygalacturonase (E.C. 3.2.1.15), xyloglucanase (E.C. 3.2.1.151), carboxylic ester hydrolase (EC 3.1.1.1), catalase-peroxidase (E.C. 1.11.1.21), and several uncharacterized enzymes.

## 4. Conclusions

Seven fungal strains were tested for their lignocellulosic activity and in all tested strains some level of hydrolytic activity (i.e., exoglucanase, endoglucanase, xylanase, mannanase, arabinose, and pectinase) required for complete hydrolysis of cellulosic material was detected. The fungi from the genus *Fusarium* showed the highest activities and a broad spectrum of produced enzymes among the tested strains. Through additional research, the results showed that both *Fusarium* strains can be used for lignocellulolytic enzyme production. Higher glucanase activities were detected in samples containing the crude enzyme of *F. oxysporum* JCM 9293 in comparison to *F. verticillioides* JCM 23107. The maximal exo- and endoglucanase activities determined were 3.33 and 0.33 U mg^−1^_total protein_ achieved after 7 and 4 days of cultivation, respectively. In contrast, *F. verticillioides* JCM 23107 showed two times higher xylanase activity than *F. oxysporum* JCM 9293. The maximal xylanase activity determined for *F. verticillioides* JCM 23107 after 7 days of cultivation was 16 U mg^−1^_total protein_.

On the basis of the obtained results, it can be concluded that both fungal strains can be used as producers of enzyme cocktails using nutrient medium 4 supplemented with corn cobs or other residual lignocellulosic feedstocks as the inducer substrate. Further research into the composition of the optimal enzyme mixture, enzyme purification, and characterization to additionally enhance the breakdown of selected pretreated lignocellulosic feedstock and to optimize the breakdown process is required.

## Figures and Tables

**Figure 1 polymers-13-03736-f001:**
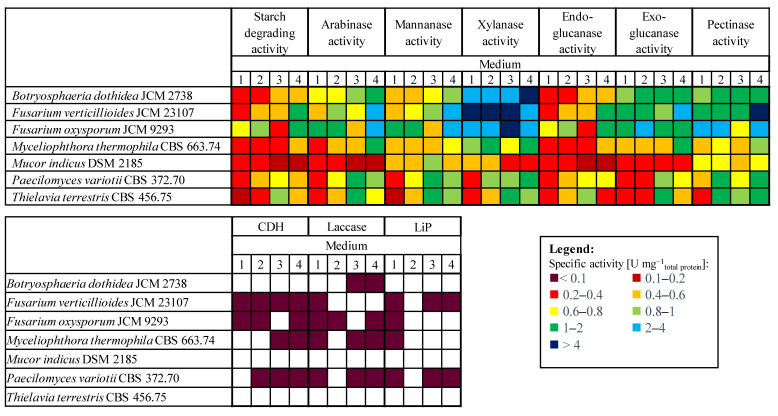
Screening of lignocellulolytic (and starch) enzyme degrading activity by different fungi in various media.

**Figure 2 polymers-13-03736-f002:**
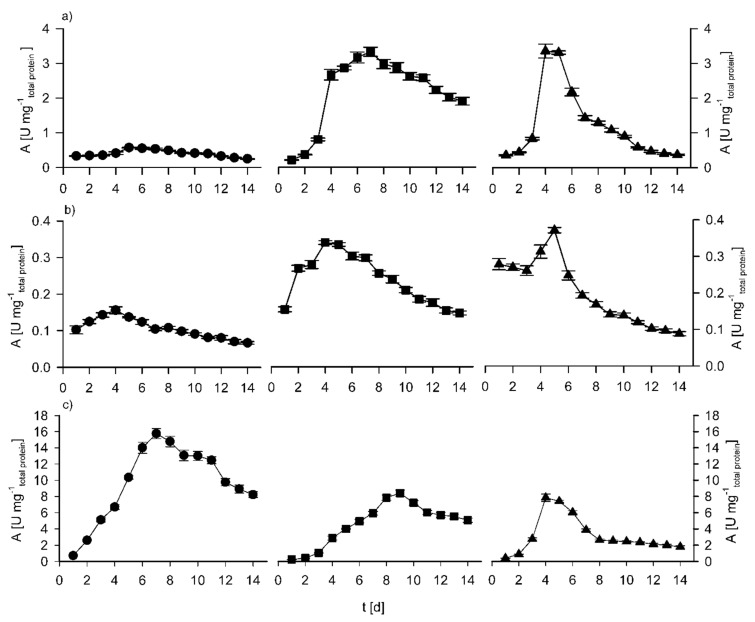
Change of enzymatic activity: (**a**) exoglucanase, (**b**) endoglucanase, and (**c**) xylanase during sequential solid state and submerged cultivation of FV (●), FO (■), and mixed culture (▲). Bars represent the standard deviation of duplicate cultivations.

**Figure 3 polymers-13-03736-f003:**
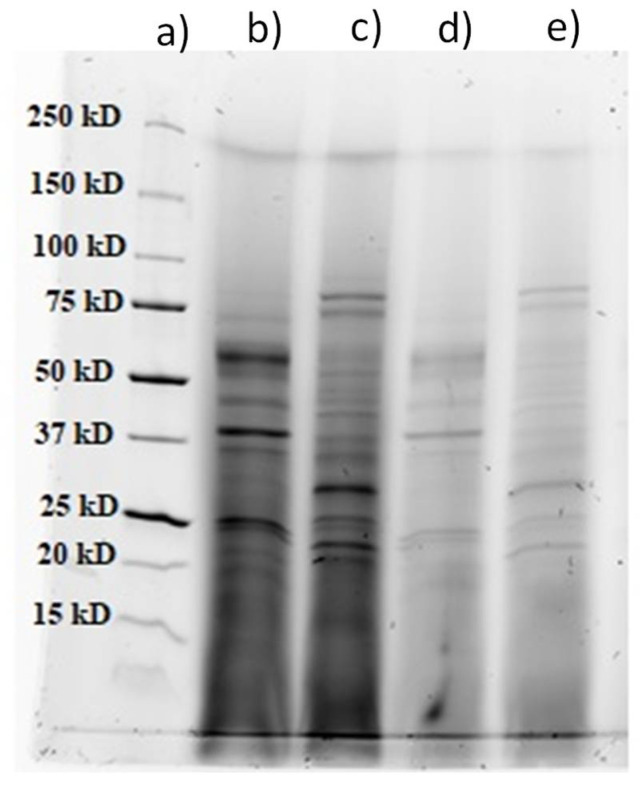
SDS-PAGE analysis of: (**a**) Precision Plus Protein Standard (Bio-Rad, USA), (**b**) partially purified extracts of *FO*, (**c**) partially purified extracts of *FV*, (**d**) crude extract of *FO*, and (**e**) crude extract of *FV*.

**Table 1 polymers-13-03736-t001:** Media composition used for cultivations of fungi.

Component (g L^−1^)	Nutrient Medium			
	1	2	3	4
Yeast extract	2	-	2	-
Peptone	5	2	5	-
Corn steep liquor	-	5	-	5
Diammonium phosphate	1.4	1.4	1.4	1.4
KH_2_PO_4_	2	2	2	2
MgSO_4_	0.2	0.2	0.2	0.2
Urea	-	-	0.3	-
CaCl_2_	-	-	0.2	-
Tween 80	-	-	1	-
MnSO_4_	-	-	-	0.01
CuSO_4_	-	-	-	0.01
NaCl	-	-	-	0.1
Salt solution * (%)	0.1	-	0.1	-

* Salt solution composition (in g L^−1^): FeSO_4·_7H_2_O, 5; MnSO_4_·H_2_O, 2; CuSO_4_, 2; ZnSO_4_·7H_2_O, 1.5; and CoCl_2_, 2.

**Table 2 polymers-13-03736-t002:** Quantification of different enzymes using DNS.

Common Name	Substrate	Released Sugar
Starch degrading	Starch	D-glucose
Xylanase	Xylan	D-xylose
Mannanase	Mannan	D-mannose
Arabinase	Arabinan	L-arabinose
Endoglucanase	Cellulose	D-glucose
Exoglucanase	Carboxymethyl cellulose	D-glucose
Pectinase	Pectin	Galacturonic acid

**Table 3 polymers-13-03736-t003:** Enzyme activities of crude and partially purified extracts of *Fusarium* sp.

	Protein Concentration [mg mL^−1^]	Total Volume [mL]	Enzyme Activity [U mg^−1^_total protein_]						
			Starch Degrading	Endo- and Exo-arabinase	Mannanase	Xylanase	Endoglucanase	Exoglucanase	Pectinase
FV crude extract	0.41 ± 0.03	980 ± 23	0.42 ± 0.08	0.58 ± 0.07	0.53 ± 0.06	14.48 ± 0.12	0.21 ± 0.01	0.57 ± 0.01	1.022 ± 0.08
FV partially purified extract	0.89 ± 0.09	132 ± 8.4	0.91 ± 0.09	1.11 ± 0.10	1.75 ± 0.10	25.12 ± 0.25	0.55 ± 0.09	0.91 ± 0.07	4.45 ± 0.11
FO crude extract	0.33 ± 0.02	965 ± 37	0.76 ± 0.07	0.36 ± 0.04	0.62 ± 0.05	6.44 ± 0.18	0.34 ± 0.02	4.09 ± 0.01	0.79 ± 0.09
FO partially purified extract	0.61 ± 0.05	120 ± 5.6	1.09 ± 0.09	0.94 ± 0.07	1.47 ± 0.10	11.05 ± 0.14	1.09 ± 0.05	7.85 ± 0.18	2.93 ± 0.23

## Data Availability

The data presented in this study are available on request from the corresponding author.

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
