# Peer review of "Screening of Lignocellulolytic Enzyme Activities in Fungal Species and Sequential Solid-State and Submerged Cultivation for the Production of Enzyme Cocktails"

_polymers, 2021, doi:10.3390/polym13213736_

Round 1

Reviewer 1 Report

This paper's aim was to provide information about seven fungal species screened for lignocellulose degrading activity. The manuscript fits within the scope of the journal. The manuscript is interesting and relatively well organized.  The title is clear and it is adequate to the content of the article.  The author’s work on discussing achieved results is appreciated but the revisions are necessary to improve the clarity of the presentation.

I have some recommendations for authors:

- Language style of the paper should be improved

- Please include in the abstract more information about the aim and results. In its present form, the abstract largely presents the experimental plan.

L99: see typo error - delete ”into”

- Please highlight in the introduction the degree of novelty and originality of the work.

- The main problem is the lack of statistical analysis. Please include a separate subchapter for Statistical analysis!

- Please structure the results and discussions on subchapters. Currently, the results are difficult to follow.

- Please improve the quality of the Figures.

Author Response

we would like to thank you for opportunity to revise our manuscript (ID: polymers-1422609). Furthermore, we would like also to give special thanks to reviewers for the insightful comments and suggestions which were very useful during revision process. In general, the whole manuscript was carefully read and corrected (major changes are in red color) according to the reviewer’s suggestions. New references are added and consequently the whole references were renumbered. Our comments to reviewer’s suggestions are presented below. 

Reviewer 1

Comments and Suggestions for Authors

This paper's aim was to provide information about seven fungal species screened for lignocellulose degrading activity. The manuscript fits within the scope of the journal. The manuscript is interesting and relatively well organized. The title is clear and it is adequate to the content of the article. The author’s work on discussing achieved results is appreciated but the revisions are necessary to improve the clarity of the presentation.

I have some recommendations for authors:

- Language style of the paper should be improved

Our comment:

We agree with this comment and consequently language style was check and improved by the experienced English speaker. 

- Please include in the abstract more information about the aim and results. In its present form, the abstract largely presents the experimental plan.

Our comment:

We agree with this comment and consequently abstract was corrected accordingly (see page 1).

L99: see typo error - delete ”into”

Our comment:

We agree with this comment and consequently correction was made accordingly.

- Please highlight in the introduction the degree of novelty and originality of the work.

Our comment:

We agree with this comment and consequently in the section Introduction the novelty and originality of this work is presented (see page 3, lines 99-107).

- The main problem is the lack of statistical analysis. Please include a separate subchapter for Statistical analysis!

Our comment:

We agree with this comment and consequently the subchapter Statistical analysis is introduced in the revised manuscript (see page 7).

- Please structure the results and discussions on subchapters. Currently, the results are difficult to follow.

Our comment:

We agree with this comment and consequently the section Results and Discussion was divided into subchapters accordingly (see pages 7-13).

- Please improve the quality of the Figures.

Our comment:

We agree with this comment and consequently the quality of figures was improved (see Figures 1-3).

We hope that current form of our manuscript is suitable for consideration to be published in journal Polymers.

Yours sincerely,

Božidar Šantek

Reviewer 2 Report

The purpose of the article is to lay the foundations for future understandings of the resource for research focused on the find fungi producing lignocellulolytic enzymes that can break down the agricultural waste into into simple sugars. The article, however, must be improved in terms of writing since some grammar and syntax errors are present in the manuscript. They should address the subject and critically review the information from the literature. There are several current works showing several microorganisms that produce lignocellulolytic enzymes present in the environment, with high biotechnological potential.

 My suggestions:

The authors need to revise the title of the paper in a more meaningful way.

The abstract is written in a way lacks logic. It should highlight the salient findings more critically.

Keywords are present in the title, choose others.

Introduction need more convincing rational for this article. 

line 42: I suggest the authors indicate promising sources for the isolation of microorganisms that produce lignocellulolytic enzymes in the environment, such as: soil, crop residues, compost... (isolation of cultivable microorganisms); as well as very important to highlight an approach of isolation of enzymes of biotechnology interest from a metagenomic approach (isolation of non-cultivable microorganisms, read and quote: Farias, Nathálya; Almeida, Isabela; Meneses, Carlos. 2018. "New Bacterial Phytase through Metagenomic Prospection" Molecules 23, no. 2: 448. https://doi.org/10.3390/molecules23020448).

The introduction has long paragraphs, I suggest reducing the size of the paragraphs.

Provide experimental work plan at the start of M&M. No detail description is available about the experiment.

What statistical method is used?

Authors should discuss the results integrally. The discussion is based on individual results. I suggest that integrating the results will give more value to the work. I suggest that you discuss by integrating all your results. You can use correlation tests (PCA or Pearson Correlation). 

The results of this study are not fully explained therefore the interpretation of the results is very difficult. The author needs to provide the % increase or decrease rather than just writing ''significantly increased….''.

Improve the legend in Figure 1.

Table 3: Please provide standard error or standard deviation of the results.

Figure 3: What type of molecular mass marker is used?

The discussion is poorly written hence, needs rewriting. The discussion should be further strengthened by adding some more relevant papers. The literature search is insufficient, only few related research papers in the past three years are cited, add the latest research results appropriately. See the below links if you think it will benefit your discussion.

Rewrite the conclusion! It needs to be much improved.

Author Response

we would like to thank you for opportunity to revise our manuscript (ID: polymers-1422609). Furthermore, we would like also to give special thanks to reviewers for the insightful comments and suggestions which were very useful during revision process. In general, the whole manuscript was carefully read and corrected (major changes are in red color) according to the reviewer’s suggestions. New references are added and consequently the whole references were renumbered. Our comments to reviewer’s suggestions are presented below. 

Reviewer 2

Comments and Suggestions for Authors

The purpose of the article is to lay the foundations for future understandings of the resource for research focused on the find fungi producing lignocellulolytic enzymes that can break down the agricultural waste into simple sugars. The article, however, must be improved in terms of writing since some grammar and syntax errors are present in the manuscript. They should address the subject and critically review the information from the literature. There are several current works showing several microorganisms that produce lignocellulolytic enzymes present in the environment, with high biotechnological potential.

Our comment:

We agree with these comments and consequently corrections in our manuscript were made accordingly (see changes through the whole manuscript).

My suggestions:

The authors need to revise the title of the paper in a more meaningful way.

The abstract is written in a way lacks logic. It should highlight the salient findings more critically.

Keywords are present in the title, choose others.

Our comment:

We agree with these comments and consequently manuscript title and abstract were corrected according to the reviewer suggestions (see manuscript title and abstract; page 1). Keywords are also changed (see page 1; section Keywords).

Introduction need more convincing rational for this article. 

line 42: I suggest the authors indicate promising sources for the isolation of microorganisms that produce lignocellulolytic enzymes in the environment, such as: soil, crop residues, compost... (isolation of cultivable microorganisms); as well as very important to highlight an approach of isolation of enzymes of biotechnology interest from a metagenomic approach (isolation of non-cultivable microorganisms, read and quote: Farias, Nathálya; Almeida, Isabela; Meneses, Carlos. 2018. "New Bacterial Phytase through Metagenomic Prospection" Molecules 23, no. 2: 448. https://doi.org/10.3390/molecules23020448).

The introduction has long paragraphs; I suggest reducing the size of the paragraphs.

Our comment:

We agree with these comments and consequently section Introduction was rewritten according to your suggestions and new references were added (see page 1-2; section Introduction).

Provide experimental work plan at the start of M&M. No detail description is available about the experiment. What statistical method is used?

Our comment:

We agree with these comments and consequently experimental setup was described in new section 2.1.. Section Statistical analysis was also introduced with description of applied method. Presented results are obtained during first stage of study related to the selection of the fungal strains that have the highest potential for lignocellulosic enzyme production. Our second study stage (current research) is focused on the improvement of enzyme cocktails composition and activities where is plan to perform PCA analysis. Current data were also used in PCA analysis which results showed that final enzyme activities is effected by a lot environmental parameters. It has to be pointed out that a lot interactions existed between these parameters and therefore further research is required to define clear correlations between them. Based on these data clear correlations between environmental parameters and enzyme activities will be possible to establish and improve the quality of final enzyme cocktails for lignocellulose degradation (see pages 3-7).

Authors should discuss the results integrally. The discussion is based on individual results. I suggest that integrating the results will give more value to the work. I suggest that you discuss by integrating all your results. You can use correlation tests (PCA or Pearson Correlation). 

The results of this study are not fully explained therefore the interpretation of the results is very difficult. The author needs to provide the % increase or decrease rather than just writing ''significantly increased….''.

Our comment:

We agree with these comments and consequently section Results and discussion was rewritten (see section Results and discussion; pages 7-13).

Improve the legend in Figure 1.

Our comment:

We agree with this comment and consequently the legend of Figure 1 was corrected and improved (see Figure 1; page 10).

Table 3: Please provide standard error or standard deviation of the results.

Our comment:

We partially agree with this comment because standard deviation of results is already presented in Table 3, but we checked again all data. We are grateful to reviewer for taking care about correctness of presented data (see Table 3).

Figure 3: What type of molecular mass marker is used?

Our comment:

We agree with this comment and consequently in the legend of Figure 3 type of molecular mass marker was added. (see Figure 3; page 13).

The discussion is poorly written hence, needs rewriting. The discussion should be further strengthened by adding some more relevant papers. The literature search is insufficient, only few related research papers in the past three years are cited, add the latest research results appropriately. See the below links if you think it will benefit your discussion.

Our comment:

We agree with this comment and consequently section Results and discussion was rewritten and new references were added (see section Results and discussion; pages 7-13).

Rewrite the conclusion! It needs to be much improved.

Our comment:

We agree with this comment and consequently section Conclusions were rewritten (see section Conclusions; pages 13-14).

We hope that current form of our manuscript is suitable for consideration to be published in journal Polymers.

Yours sincerely,

Božidar Šantek

Round 2

Reviewer 1 Report

The authors made appropriate additions and corrections postulated by the Reviewer. Currently, I have no objections to the submitted work.

Reviewer 2 Report

Thanks for attending all the suggestions. The manuscript has been significantly improved. I consider that the work has enough quality to be considered for publication in Polymers (MDPI).